# Sessō Sōsai and the Chinese Anti-Christian Discourse

Qiaoyu Han

Graduate School of Interdisciplinary Information Studies, The University of Tokyo, Tokyo 113-0033, Japan; han-qiaoyu766@g.ecc.u-tokyo.ac.jp

**Abstract:** The early Tokugawa period witnessed the establishment of anti-Christian policy as a significant agenda. In 1647, Sessō Sōsai, a Zen monk, undertook the task of delivering sermons in Nagasaki, aiming to convert the local population to Buddhism. Following his preaching, Sessō authored two anti-Christian texts, with the second text reflecting a pronounced influence from Chinese Buddhist anti-Christian discourse. This article seeks to explore the correlation between Sessō's anti-Christian writings and his engagement with the Chinese Buddhist community in Nagasaki. By delving into the analysis of personal networks, this study illustrates Sessō's familiarity with the evolution of Buddhism in China and his incorporation of ideas from the Chinese Buddhist anti-Christian movement during his time.

**Keywords:** Sessō Sōsai; Feiyin Tongrong; Miyun Yuanwu; Buddhism; Kirishitan; anti-Christian treaties; Huangbo; Nagasaki; cultural exchange





## 1. Introduction

The early Tokugawa period (1603–1867) witnessed a significant shift in the Tokugawa shogunate's attitude towards Catholic missionaries. Initially, Tokugawa Ieyasu 徳川家康 (1543–1616) displayed generosity by offering substantial financial aid to the Jesuit Society upon learning that a Portuguese ship had been seized by the Dutch in 1603 (Rodrigues 2003, p. 112).[1] However, a mere 10 years later, in 1613, Ieyasu abruptly issued an edict known as the *Bateren tsuihō no fumi* 伴天連追放之文, which resulted in the expulsion of the missionaries and the subsequent underground nature of missionary activities. Despite the forced departure of the majority of missionaries from Japan, the Christian communities remained sizeable, comprising hidden Christians known as *kakure kirishitan* 隠れキリシタン. Alongside the ban on Christianity, the shogunate introduced the *danka* system 檀家制度, which assigned the responsibility of verifying the orthodoxy of parishioners to Buddhist temples. Individuals were required to officially register as Buddhist believers within the temple. In 1637, a notable uprising known as the Shimabara Rebellion 島原の乱 occurred in the Shimabara Peninsula and Amakusa. This rebellion stands as the most significant civil unrest during the entire Tokugawa period, resulting in the estimated death of over 20,000 rebels (Nakamura 1988, p. 165; Gonoi 1990, p. 223),[2] many of whom were hidden Christians.

The Shimabara Rebellion had a profound impact on the Tokugawa shogunate's control over religious affairs, particularly in the Kyushu region. In 1640, the shogunate implemented the *shūmon'aratame* system 宗門改制度, with Inoue Masashige 井上政重 (1585–1661) assuming the role of Head Investigator 宗門改役 (Ohashi 2019, pp. 46–54). This system aimed to identify and convert hidden Christians throughout the archipelago into officially recognized religions. Despite the shogunate's anti-Christian policies, Christianity and Portuguese influence persisted. In 1642 and 1643, the Jesuits made two unsuccessful attempts to send missionaries (known as the Rubino groups) to Japan, in an effort to support the mission and establish contact with Christovão Ferreira (1580–1650), an apostate Jesuit Vice-Provincial of Japan who was captured by the authorities in 1633 and subjected to torture before being coerced into converting to Buddhism (Cieslik 1974). In July 1647,

a Portuguese embassy arrived in Nagasaki, seeking to reopen diplomatic relations, but it was promptly ordered to leave. These European endeavors only further solidified the shogunate's belief that Japan remained vulnerable to Catholic infiltration.

In such circumstances, under the directive of the shogunate, the monk Sessō Sōsai 雪窓宗崔 (1589–1649) proceeded to Kōfukuji 興福寺 in Nagasaki. This temple, constructed by Chinese residents in Nagasaki, served as the location where Sessō delivered sermons to the public shortly before the arrival of the Portuguese embassy in June 1647 (Okuwa 1984, p. 306).[3] Subsequently, Sessō compiled a written account known as *Kōfukuji hikki* 興福寺筆記, summarizing the key points from his sermons in Nagasaki. The following year, Sessō departed Nagasaki and returned to Tafukuji 多福寺 in Usuki 臼杵, where he held the position of abbot. During his time at Tafukuji, Sessō authored another anti-Christian work titled *Taiji jashū ron* 対治邪執論. It is likely that Sessō acquired a deeper understanding of Christianity subsequent to his preaching, as *Taiji jashū ron* presents a more comprehensive critique of the religion rather than a mere reproduction of his earlier sermons.

While several scholars have asserted that the predominant essence of the suppression directed against Christianity during the early Tokugawa period was primarily socio-political rather than rooted in philosophical or religious motivations (Paramore 2009; Hur 2007; Ōhashi 1996), the involvement of individuals aiding the shogunate in executing anti-Christian persecution was not confined solely to socio-political rationales. Indeed, Sessō's compositions opposing Christianity, particularly his later opuses, stand as notable instances of integrating intricate philosophical and religious condemnations of Christian doctrines into the discourse of early Tokugawa's anti-Christian campaign. Possibly stemming from the emphasis placed on the socio-political dimension of the early Tokugawa anti-Christian suppression, analyses pertaining to Sessō's religious scrutiny of Christianity in English academic discourse are primarily contained within Ikuo Higashibaba's exploration of Christianity and Jan C. Leuchtenberger's investigation of Kirishitan literature during the Early Modern Japan era (Higashibaba 2001; Leuchtenberger 2013).[4] However, a dialogue concerning his philosophical concepts remains absent in the realm of the English literature. The objective of this article is to undertake an analysis of Sessō's *Taiji jashū ron* while exploring the Chinese origins of several pivotal philosophical critiques articulated therein.

The study of *Taiji jashū ron* and Sessō in general commenced several decades ago. However, our understanding of Sessō remained severely limited until the mid-1980s due to a dearth of primary sources. Fortunately, the groundbreaking efforts of Okuwa Hitoshi (1937–2020) and his colleagues shed new light on the subject. They transcribed a collection of documents preserved in Tafukuji pertaining to Christianity, enabling us to analyze Sessō's anti-Christian writings using these newfound materials.[5] Furthermore, it is worth noting that Okuwa has already identified numerous connections between Sessō and Chinese monks in Nagasaki within his research. Regrettably, he overlooked certain sections in which Sessō directly drew upon Chinese anti-Christian texts, consequently neglecting an examination of the transmission of Chinese anti-Christian literature from China to Nagasaki.

More recently, Nishimura Ryo (1972–2016) conducted research on Sessō's critique of Christianity, drawing heavily on Okuwa's work. In her study, she highlighted an overlooked yet significant connection between Sessō and Chinese Buddhist monks, particularly the Zen monk Feiyin Tongrong 費隱通容 (1593–1661), whose anti-Christian text, *Yuandao pixie shuo* 原道闢邪說 (1636), was directly quoted in *Taiji jashū ron* (Nishimura 2011).

Additionally, building upon Okuwa's research, Martin Nogueira Ramos recently undertook an analysis of Sessō's writings from a political standpoint, focusing on the Tokugawa anti-Christian policy and the ramifications of the Shimabara Rebellion (Nogueira Ramos 2020). His essay also provides insights into the content of the three documents preserved in Tafukuji.



Nishimura's and Nogueira Ramos's research endeavors offer valuable insights into the implications of Sessō's anti-Christian writings. However, neither author has thoroughly examined the connection between Feiyin and Sessō in terms of their anti-Christian ideologies. While Nogueira Ramos briefly acknowledges Nishimura's work, his focus on the political aspects prevents him from delving into Feiyin Tongrong's philosophical ideas. On the other hand, Nishimura identifies and analyzes Sessō's quotation from Feiyin within a Buddhist anti-Christian framework. She suggests that Sessō borrowed the concept of the Great Way of the Void 虚空の大道 from Feiyin as a metaphysical and ontological reality.

If we rely solely on Sessō's quotation, it may lead us to believe that Sessō and Feiyin both understood the Great Way of the Void as synonymous with the Buddhist concept of emptiness 空. However, a meticulous examination of Feiyin's original text reveals that his argument is rooted in a syncretic fusion of Buddhism and Confucianism. Interestingly, Sessō intentionally omits all the Confucian elements in his extensive quotation of Feiyin's text. This revision of the original text, which was not addressed by Nishimura, warrants deeper investigation. Thus suggests that Sessō's engagement with Feiyin's arguments was not a passive acceptance, nor did he solely rely on Feiyin's textual content, as Nishimura's analysis might inadvertently imply. Furthermore, this revision highlights the necessity of harmonizing a Chinese text with the socio-political and intellectual milieu of early Tokugawa Japan in some cases.

In addition to the absence of a thorough comparison between Feiyin's original text and Sessō's quotation, the originality of Feiyin's work has been inadvertently disregarded by Nishimura. Through a meticulous examination of *Biantian shuo* 辯天說 (1635) (Zhong and Xu 1984), authored by Feiyin's master Miyun Yuanwu 密雲圓悟 (1567–1642), it becomes evident that the anti-Christian argument quoted by Sessō bears striking resemblance to the discourse presented in *Biantian shuo*, which was published one year prior to Feiyin's *Yuandao pixie shuo* that Sessō drew upon. Nishimura acknowledges this similarity in her article discussing the development of Chinese Buddhist anti-Christian discourses during the Late Ming period. However, she does not explore this aspect in her article specifically addressing Sessō's incorporation of Feiyin's ideas (Nishimura 2010). Therefore, it is imperative to consider that Sessō's utilization of Feiyin's concepts should not be hastily interpreted as mere individual quotation, but rather necessitates evaluation from a collective standpoint.

In summary, while Nishimura has presented compelling evidence that underscores Sessō's substantial reliance on Chinese sources within his anti-Christian compositions, there remains a certain ambiguity concerning the specific modifications he applied to the quoted texts. Moreover, the precise nature of Sessō's interactions with the Chinese Buddhist community, which culminated in his utilization of Chinese textual material, continues to elude comprehensive understanding.

This essay endeavors to undertake a re-evaluation of the impact exerted by Chinese monks' anti-Christian writings on Sessō's texts, focusing particularly on Sessō's discerning adoption of Feiyin's anti-Christian ideology, while also extending the examination to include Miyun's anti-Christian writing. The intention is to showcase that, beyond the three Tafukuji documents, Chinese sources significantly influenced Sessō's anti-Christian discourse especially in the philosophical arguments. By conducting an interpersonal analysis of the Buddhist communities in China and Japan, the objective is to elucidate the intricate network of individuals that facilitated the transmission of anti-Christian discourses from China to Japan, specifically among Zen monks. Moreover, it seeks to introduce an alternative angle to the examination of the early Tokugawa anti-Christian persecution. This perspective entails situating the anti-Christian initiatives undertaken by figures such as Sessō within the framework of their personal aspirations. The objective is to explore how these individuals could leverage their fervent ambitions to effectively fulfill their official duties, thereby accomplishing their obligations while simultaneously benefiting from their deep-seated passions.

## 2. A Brief Biography of Sessō

Sessō Sosai was born in Bungo 豊後 in 1589.[6] He developed a profound interest in Buddhism at a young age and subsequently entered the Shinshōji temple 真正寺 of the Pure Land sect in 1599. Following his studies at Shinshōji and Zenshōji temple 善正寺, Sessō grew dissatisfied with the practice of *nenbutsu* 念仏[7] and sought to delve into the realm of Zen philosophy. At the age of 25, Sessō joined Tafukuji, an esteemed temple affiliated with the Myōshinji branch 妙心寺派 of Rinzai Zen 臨済, eventually rising to the position of abbot.

In 1616, at the age of 28, Sessō journeyed to Edo, where he forged affiliations with several prominent monks of his era. Noteworthy among these individuals were Gudō Toshoku 愚堂東寔 (1577–1661), Daigu Sōchiku 大愚宗築 (1584–1669), Ungo Kiyō 雲居希膺 (1582–1659), and Ryōdō Sōketsu 了堂宗歇 (1587–1664), all of whom were affiliated with the Myōshinji branch. Additionally, he established connections with Suzuki Shōsan 鈴木正三 (1579–1655) and Bannan Eishu 万安英種 (1591–1654), notable figures within the Sōtō sect. Sessō joined forces with these monks in their collective endeavor to reinvigorate Zen Buddhism in Japan. Notably, Suzuki Shōsan 鈴木正三 (1579–1655), a key member of this progressive cohort, authored an influential anti-Christian treatise titled *Ha Kirishitan* 破切支丹 in 1642, as a direct response to the Shimabara Rebellion (Okuwa 1984, p. 292).

Not only did Sessō befriend numerous influential monks of his era, but he also progressively gained renown among the daimyo in Edo. Among these dignitaries, Inaba Kazumichi 稲葉一通 (1587–1641), the daimyo of the Usuki domain, encountered Sessō at Tōzenji 東禅寺 in Edo and subsequently assumed the role of Sessō's disciple. Kazumichi was captivated by Sessō's exceptional talents, prompting him to extend an invitation for Sessō to undertake the reconstruction of Tafukuji in Usuki.[8]

As Sessō's reputation continued to swell, he embarked on a journey to Kyoto in 1633, ascending to the esteemed position of the First Seat of Myōshinji, a position of considerable significance within the Myōshinji branch. Sessō's dynamic presence in Kyoto ultimately facilitated a remarkable opportunity to elucidate Zen kōan to Emperor Go-Mizunoo 後水尾天皇 (1596–1680) in the year 1639. This auspicious encounter culminated in the conferral of the venerable purple robe 紫衣 upon Sessō, emblematic of his exceptional spiritual attainments. This honor was bestowed just a year prior to Sessō's delivery of his anti-Christian sermons in Nagasaki.

Based on Okuwa's conjecture, it is suggested that Sessō encountered Suzuki Shōsan in Edo in 1647. Through Shōsan's endorsement, Sessō was dispatched to Nagasaki by Hoshina Masayuki 保科正之 (1611–1673), an intimate associate of Shōsan and a prominent figure within the shogunate's upper echelons (Okuwa 1984, pp. 304–5). The precise rationale behind Sessō's designation to deliver anti-Christian sermons remains unrecorded. Nonetheless, it is plausible to surmise that due to his proximity to Suzuki Shōsan, who had previously undertaken endeavors against Christianity in the Kyushu region a few years earlier, Sessō likely harbored a shared aspiration with Shōsan to rejuvenate Buddhism within a region influenced by Christianity. This mutual objective seemingly positioned Sessō as a suitable candidate for this undertaking.

Owing to the scarcity of resources, there remains ambiguity regarding whether Sessō possessed any prior knowledge of Christianity or engaged in anti-Christian endeavors prior to his sermons in Nagasaki. However, as confirmed by the biographical record of the first three abbots of Tafukuji, the *Tafukuji gyōyū* 多福寺行由, it is clear that he was dispatched by a prominent shogunate official 宰官 (Okuwa 1984, p. 20). Sessō arrived in Nagasaki during the summer of 1647 and proceeded to deliver four sermons to the local population within a span of twenty-one days. According to the account in *Kōfukuji hikki*, a total of 22,820 individuals received the Buddhist Five Precepts 五戒 following the completion of Sessō's sermons. However, Kengan Zenetsu 賢巌禅悦 (1618–1699), in his two biographies of Sessō, reported a slightly lower figure of around 18,000. This notable achievement was hailed as a significant success by Inoue Masashige, who arrived in Na-

gasaki shortly after the conclusion of the sermons and relayed the news to the shogun Tokugawa Iemitsu 德川家光 (1604–1651).

After the preaching, Sessō remained in Nagasaki for a period of time and completed his work on *Kōfukuji hikki*. The following year, he returned to Tafukuji and composed *Taiji jashū ron*, only to pass away in the subsequent year.

### 3. The Structure of *Taiji jashū ron*

*Taiji jashū ron* is not a straightforward anti-Christian text, as it encompasses a thorough and multifaceted critique of Christianity. The text can be dissected into three distinct parts, each containing an exposition of Kirishitan teachings and Sessō's accompanying commentary on them.

The first part of the discussion pertains to the historical progression of Christianity in Japan and its resemblance to Buddhist sects such as Pure Land and Nichiren. According to Sessō, the initial cohort of European padres (頗姪連)[9] to set foot on the shores of Bungo consisted of Saint Francis Xavier (三跗乱志須古娑毘恵娑, 1506–1552) and 我須頗娑 (likely Gaspar Vilela, 1526–1572, who arrived in Bungo in 1556) during the final year of Tenbun 天文 (Jan 1555–Jan 1556) from Rome (浪魔).[10] The missionaries employed a strategy aimed at enticing individuals to attend their private catechism sessions for a duration of seven days (堅鎖門戸而不令他人聞法理，潜説法要者一七箇日). Sessō posited that Jesus (是寸須), the figure revered by the missionaries, might have embraced Buddhism (帰依釈氏), but only acquired a superficial understanding of its concepts and rituals (而学名相). Having grasped merely the surface of Buddhist teachings, Jesus amalgamated elements of the doctrines propagated by the Six Heretical Teachers (*rokushi gedō* 六師外道) with Buddhist terminology and rituals, resulting in the formulation of a novel heretical doctrine (作外道邪見).

In the second part, Sessō delves into the teachings imparted to newly converted Kirishitan on their first day. The discussion commences with a Christian critique of Buddhism, asserting its inherent emptiness and voidness. Specifically, three Buddhist sects—Nichiren, Pure Land, and Zen—are singled out, alongside Shinto. The Jesuits contend that adherents of Nichiren and Pure Land, rather than placing their faith in the Christian God, blindly bestow their devotion upon Buddha Shakyamuni and Amida, respectively, whom the Jesuits perceive as mere mortals. Regarding Zen, the missionaries perceive it as fundamentally devoid of substance. Lastly, Shinto becomes the target of criticism. The missionaries view the kami 神 worshipped by the Japanese as the spirits of deceased humans or animals (祀其霊以為神). They deride the notion that human and animal spirits can exert influence on the physical world and denounce it as heretical.

In response to the aforementioned catechism on the first day, Sessō persistently maintains that Jesus failed to comprehend the truth of Buddhism but instead sought a Creator beyond this worldly realm. Drawing from a Buddhist metaphysical standpoint, he refutes this idea and proposes that the three realms of existence are purely manifestations of the mind, and the myriad phenomena are but manifestations of consciousness (三界唯心，万法唯識). Furthermore, he cites an ancient sage (古徳) who proclaimed, "Heaven and Earth share the same root, and all things are interconnected as one entity (天地同根, 万物一体)." Sessō, in accordance with Buddhist cosmology, regards all phenomena as fundamentally united within a singular generative entity. This quotation can also be found in Feiyin's *Yuandao pixie shuo*, where the exact phrase (天地同根, 万物一体) is employed to challenge the Christian notion that seeks to establish an external Lord of Heaven, divorced from the realm of the mind (亦背一心之道 … 妄執有一天主)[11].

Proceeding to the third part, we encounter a notable presence of direct Chinese influence. Sessō initiates this segment by providing a synopsis of the Genesis narrative, encompassing God's creation of the world up until the moment of Adam and Eve's fall. Subsequently, an elucidation of the concept of Original Sin and the redemptive significance of Jesus' Passion in the salvation of humanity ensues. Additionally, the manner in which baptism was administered and the steadfastness displayed by individuals in their adher-

ence to their faith through martyrdom are expounded upon. In his commentary on these matters, Sessō initially raises a moral paradox:

> Deus … creates Heaven and Hell, sorrowness and happiness, and make people suffer [from pain and sorrowness]. How could this be described as the source of wisdom and compassion? You said, the Lord of Heaven (天主) would cancel one's sin and elevate that person to Heaven however great the sin is as long as that person possesses the faith. For those who do not possess faith, even if they are sages or saints, they fell into inferno. Can this be the source of the Law (憲法)? The vast world (大千世界) includes countless range of territory, only people living in the western part have faith in Deus, the rest of the people in other countries have not heard Deus before, how can they put faith in Deus? However, these people being the creations of Deus are not given a chance to hear from Him. Does this deserve the title of the lord of all virtues? (Okuwa 1984, p. 101)

This moral quandary finds its origins in the writings of various anti-Christian authors during the early seventeenth century. In Japan, Christovão Ferreira, a former Portuguese Jesuit priest who adopted the name Sawano Chūan 沢野忠庵 (1580–1650) following his apostasy in 1633, presented a similar critique in his anti-Christian treatises Gengi roku 顕偽録 (1636). George Elison, in his examination of anti-Christian treatises from the early Tokugawa period, suggests a strong likelihood of Sessō drawing heavily from Sawano's work, perhaps even engaging in direct discourse with him (Elison 1973, pp. 231–32). Elison's conjecture is reasonable and plausible, given that Sawano was stationed in Nagasaki, serving as an interpreter and assistant for the shogunate's anti-Christian inquisition (Cieslik 1974, pp. 26–30). However, there exists another potential source of anti-Christian discourse that Sessō could have drawn upon—Chinese anti-Christian texts. Indeed, numerous Chinese authors in the 1630s and 1640s employed similar lines of reasoning. For instance, Shi Ruchun 釋如純 (date unknown), in his work Tianxue chupi 天學初闢[12], questioned the omnipotence and omniscience of God, as well as the existence of evil, in relation to the fall of Adam and Eve and humanity's Original Sin. Similarly, the esteemed monk Ouyi Zhixu 蕅益智旭 (1599–1655) expressed similar sentiments in his Pixieji 闢邪集[13] (preface dating to 1643). After engaging in this moral discussion, Sessō proceeds to provide an extensive quotation from Feiyin.

## 4. The Long Quotation

We shall now direct our focus towards Sessō's citation of Feiyin Tongrong's *Yuandao pixie shuo* within the pages of *Taiji jashū ron*. The complete passage, along with its translation, can be found in Appendix A. It is worth noting that, while Sessō only made minor modifications to the original text in his citation, it is intriguing to observe that he occasionally omitted certain portions of sentences and reorganized the order of Feiyin's words. In light of Sessō's alterations, I have divided his text into 10 distinct blocks, each corresponding to a particular section of Feiyin's original composition.

When comparing Sessō's text to Feiyin's, several noteworthy points arise for discussion. Firstly, in Sessō's discourse, his presumed opponent in the debate is Christ (喜利志徒), whereas, in Feiyin's text, the focal point of contention is Matteo Ricci 利瑪竇 (1552–1610) (Higashibaba 2001, pp. 85–86). Consequently, it becomes imperative to examine Sessō's comprehension of Ricci and the texts authored by Ricci, particularly *Tianzhu shiyi* 天主實義, which evidently constitutes the target of Feiyin's refutation in his treatise.

Ricci's *Tianzhu shiyi* gained popularity among the Jesuits in Japan shortly after its publication in 1603. The Jesuit Visitor Alessandro Valignano (1539–1606) even commissioned a reprint of *Tianzhu shiyi* in Guangdong 広東 explicitly for the purpose of exporting copies to Japan to meet the demand in 1605 (Ricci 2018, p. 316). Additionally, Japanese scholars outside the Jesuit Society were also familiar with *Tianzhu shiyi*. Hayashi Razan 林羅山 (1583–1657), for example, noted in a record of his debate with Fucan Fabian 不干斎巴鼻庵 (1565–1621) in 1606 that he had read *Tianzhu shiyi* and found it to be absurd.[14] Despite the

shogunate's issuance of the ban on Christian books 禁書令in 1630 and the establishment of a system to scrutinize the content of imported Chinese books to ensure they were not related to Christianity (Ito 1936a, 1936b), Ricci's works continued to circulate clandestinely in Japan. It is plausible that Sessō, as an anti-Christian preacher dispatched by the government, could have obtained access to Ricci's books with the assistance of the authorities.

In *Kōfukuji hikki*, there is no indication that Sessō had already read *Tianzhu shiyi* during the time he delivered his sermons in Nagasaki. Therefore, if he did come across *Tianzhu shiyi*, it would have likely been after his sermons, and there are two main potential sources for this: the Chinese monks in Nagasaki or Inoue Masashige, who arrived in Nagasaki shortly after Sessō concluded his sermons. Regarding the first possibility, Sessō did interact with the Chinese Buddhist community in Nagasaki, and we delve into his connection with them later. As for the latter possibility, Tanihata Akio has conducted an extensive examination of the available source materials accessible to Sessō (Okuwa 1984, pp. 410–11). He suggests that Sessō's understanding of Christian doctrine in *Kōfukuji hikki* was surprisingly rudimentary compared to that in *Taiji jashū ron*. This is attributed to the fact that Sessō was hastily dispatched by the shogunate to deliver sermons in Nagasaki, leaving him with limited time to thoroughly study the Christian doctrine before completing his mission. Hence, Inoue, in his capacity as the Head Inspector, with easy access to various Christian-related materials, including Chinese catechisms, likely provided additional resources to Sessō. However, despite an improvement in his understanding of Christian doctrine evident in *Taiji jashū ron*, similar to *Kōfukuji hikki*, no definitive textual evidence can be found to suggest that Sessō had read *Tianzhu shiyi* or any of Ricci's works. In the absence of new evidence, it is prudent to propose that Sessō may have heard of Ricci's name but had not personally read the latter's texts.

Hence, proceeding from the assumption that Sessō had not read *Tianzhu shiyi*, the question remains: how did Sessō perceive Matteo Ricci in *Yuandao pixie shuo*? Given that Sessō replaced Ricci with Christ, let us first examine Sessō's perception of Christ. In Sessō's view, Christ was regarded as a heretical Buddhist teacher who failed to grasp the authentic Buddhist doctrine, but merely acquired superficial knowledge of its names and forms. Sessō consistently accused Christ of fabricating conceptual notions, disregarding true teachings, and introducing heretical ideas into the Buddhist doctrine. According to Sessō, Christ blended elements of Buddhist teachings with the Six Heretical teachings, resulting in the formation of the Christian doctrine. This represents a departure from Feiyin's understanding of the development of Christian doctrine. In *Tianzhu shiyi*, there is no explicit mention of Jesus Christ as the creator of Christian doctrine. Instead, Ricci establishes the existence of God and expounds upon Christian doctrines from a scholastic philosophical perspective. Naturally, Feiyin targeted Ricci for refutation because Jesus is only briefly mentioned at the end of *Tianzhu shiyi* as a savior, rather than a teacher. Since Ricci's teachings closely align with Sessō's understanding of Christ's teachings, Feiyin's refutation of Ricci can effectively serve as a means to refute Christ's teachings in *Taiji jashū ron*. Furthermore, and perhaps of greater significance, there was no compelling reason for Sessō to complicate the narrative by introducing another figure—Ricci—who had minimal relevance to the Kirishitan issue in 1640s Japan. The 1630 prohibition on Chinese Christian books explicitly identifies "32 books composed by the European Matteo Ricci and his colleagues"[15] as containing heretical teachings that should be banned. Consequently, the name Matteo Ricci had already become a sensitive matter after 1630, and referencing him could potentially give rise to unnecessary complications. Hence, Sessō's decision to omit any mention of Ricci appears to be a reasoned choice.

Sessō not only altered the name, but also rearranged Feiyin's original text. I have labelled the 10 sections as (1) to (10), and, if they were listed according to Feiyin's text, the order would be (1), (2), (3), (4), (5), (7), (8), (6), (10), (9). The initial five sections of the quotation align with the original text, but omit four sections of text between the quotes.[16] When comparing them to *Yuandao pixie shuo*, sections (1) and (2) in Feiyin's original text serve as an introduction. Feiyin commences his argument by asserting that the origin of

creation, which is without beginning or end, is not God but rather the Great Way (大道) that is inherent in every human being. Concerning the metaphysical essence of the Great Way, Feiyin states:

> Unbeknownst to you, this beginninglessness and endlessness is indeed the origin of the Great Way, and also the purpose of the ultimate truth. Moreover, this ultimate truth is within everyone, and the origin of the Great Way is not absent in anyone. It does not increase in the saints, nor decrease in the ordinary people. It exists in the sky as the sky, in humans as humans. As for all things, so it is, and for all phenomenon, it is also the same. Thus, there is no duality, no division, no distinction.[17]

Hence, if all things and phenomena inherit the Great Way and the ultimate truth, which are without a beginning or end, it becomes unnecessary and even ontologically illogical to posit the existence of a God who creates beings such as humans and other entities that manifest clear beginnings.

This is precisely the point of contention that both Feiyin and Sessō identify in the Christian doctrine. In the preceding sections of *Taiji jashū ron*, Sessō has already contended that Jesus succumbed to heretical teachings due to his limited grasp of Buddhism, having merely acquired superficial knowledge of its terminology and rituals without comprehending its underlying truths. Sessō's criticisms of Christianity primarily revolve around how Kirishitan practices imitate or modify Buddhist customs and concepts. These reproaches predominantly target the practical aspects of the religion. However, in Sessō's quotation of Feiyin's work, he expounds upon the more fundamental philosophical reason that gave rise to Jesus's heretical perspective, employing Feiyin's words as a means of elucidation:

> Christ do not understand this concept [the Great Way] and uses his mind and consciousness to explore and estimate everything within heaven and earth. When he reaches the vast and mysterious depths, he cannot quite grasp it. Therefore, he arbitrarily assumes the existence of a God, an entity with no beginning and no end, capable of giving birth to heaven and earth, and fostering all beings. Further, [he teaches that] all these beings indeed have a beginning and an end, such as birds, animals, plants, and trees. And there are entities that have a beginning but no end, such as the earth, heaven, ghosts, gods, and human souls. Only God has no beginning and no end, capable of creating everything.

In this instance, Sessō explicitly highlights the error committed by Jesus: the invention of God as the sole creative force in place of the Great Way. Jesus made this mistake due to his inability to grasp the true concept of the Great Way. Nishimura argues that the notion of the Great Way, which Sessō draws upon, serves as the underlying metaphysical foundation as his cosmological and metaphysical basis (Nishimura 2011, p. 89). However, this argument regarding Jesus's lack of understanding regarding the Great Way can be further traced back to Feiyin's teacher, Miyun, and his work *Biantian shuo*:

> You, being unable to grasp the essence of the Great Way, merely pursue names and forms (但逐名相). Therefore, you cling to Tianzhu 天主 as the Lord of Heaven, Buddha as Buddha, and sentient beings as sentient beings. And you do not understand the Buddha are those who attain awakening, and those who attain awakening are those who realize. If every person awakens and realizes, then every person is a Buddha. So why make distinctions between the heaven, human, and other living beings? Therefore, the Buddha has no fixed form. In sky, it manifests as sky; in a human, it manifests as a human. It cannot be perceived through appearance or sought through sound. It is because it is inherent in everyone complete and fulfilled. (Zhong and Xu 1984, pp. 11518–19)

Further, in part (1), Feiyin concluded the following:



The origin of the Great Way is clearly visible, without that or this. The body of the ultimate truth is beginningless and endless, the Way is equal, and hence a great harmony is formed (一道平等而浩然大均矣).

A similar argument also appears in Miyun's text:

Therefore, our great sage teaches that all sentient beings possess the wisdom and virtues of a Tathagata. It is the personal realization of the Great Path of Unity (大通之道). Through personally realizing the Great Path of Unity, one does not perceive the distinctions of superiority or inferiority between oneself and others. Thus, the Way is equal, and hence a great harmony is formed (一道平等，浩然大均矣). (Zhong and Xu 1984, p. 11521)

Miyun completed his *Biantian shuo* a year prior to Feiyin's *Yuandao pixie shuo* in 1635, indicating that Feiyin likely engaged in discussions with Miyun regarding the errors in Christian doctrines and further expanded upon the notion of the Great Way in his treatise. It is evident that Miyun's arguments bear a striking resemblance to those presented by Feiyin and Sessō. The Great Path of Unity discussed in Miyun's text corresponds to the concept of the Great Way found in Feiyin's work. Both concepts are rooted in Buddhism and emphasize the all-encompassing nature of the Buddha-nature as the singular origin of the diverse phenomena. Thus far, it appears that Sessō drew upon Miyun's and Feiyin's Buddhist metaphysics and cosmology as a counterpoint to the Christian doctrine of Creation. However, upon examining the omitted sections between parts (3) and (4), we discover that the Chinese masters' conceptualization of the Great Way differed significantly from Sessō's emphasis.

In Feiyin's original text, parts (1) and (2) serve as an introduction, while parts (3) and (4) present the central arguments to substantiate the claim that the Great Way, which has no beginning or end, pervades all things. The omitted sections indicate that Feiyin emphasizes two distinct approaches to support this assertion, whereas Sessō combines them into a single argument. Additionally, between sections (3) and (4), Sessō omits a crucial part of Feiyin's text that references the Confucian classics:

Again, "What does Heaven say? It moves with the four seasons, giving birth to all things." Thus, Heaven does not possess a conscious mind for discrimination, enabling it to follow the four seasons and give birth to all things. It harmonizes with the four seasons and all things without any deficiencies or contradictions. Furthermore, "The virtues of ghosts and gods are truly magnificent! They cannot be seen when looked at, nor heard when listened to, yet they encompass all things." Since ghosts and gods cannot be perceived through sight or hearing, yet they can encompass all things, it can be said that ghosts and gods also lack a conscious mind for discrimination, and their virtues are indeed magnificent. Moreover, Confucius extols the virtues of ghosts and gods to such a degree, but Matto claims they have beginning but no end, can it be considered appropriate?[18]

It becomes evident that Feiyin utilizes references to the Confucian classics as evidence to support his argument. When he states, "Therefore, the sutra says, 'All phenomena do not arise by themselves, nor are they born of others. They are neither conjoint nor without cause. Therefore, it is said they are unborn,'" the teaching of the sutra corresponds to Confucian philosophy. This indicates an unmistakable effort to synthesize Confucianism and Buddhism within Feiyin's text (Wu 2018).[19] In the case of Sessō, however, he omits the entirety of the Confucian references, thus creating the impression that the sutra is quoted solely to demonstrate that all phenomena are without a beginning or end and can be confirmed through human testimony from a Buddhist standpoint.

In addition to the Confucian part before part (4), Feiyin in the original text also referred to other important Confucian philosophers' teachings. Before Sessō's part (5), we find this inclusion:



Mencius said, "All things are within me." Zisi said, "Achieving harmony, Heaven and Earth find their places, and all things are nurtured." Chengzi said "When expanded, it fills the universe; when contracted, it retreats and conceals within secrecy." The sutra also states, "When the mind arises, various phenomena arise; when the mind ceases, various phenomena cease."[20]

It is important to note that, in the original text, parts (4) and (5) are not connected. In parts (3) and (4), Feiyin's original text explores two different approaches to demonstrate the presence of the beginningless and endless Great Way within all things. In part (5), the original text delves into the concept of "All dharmas converge into mind (萬法會歸唯心)", a topic that Sessō had already addressed in a previous section of his treatise.[21] Once again, prior to expounding his argument in Buddhist terminology, Feiyin chose to reference Confucian classics in order to bolster his position and promote the syncretism of Confucianism and Buddhism. In Sessō's text, not only is the Confucian component omitted, but the Buddhist quotation from the sutra of the *Awakening of Faith in the Mahayana* 大乘起信論, which Feiyin employed to align with the Confucian classics, is also omitted.

Why did Sessō choose to omit the Confucian sections in Feiyin's text? Nogueira Ramos, in his study of Sessō, suggests that *Taiji jiashū ron* was primarily intended as a guidebook for a small circle of monks close to Sessō. He hypothesizes that the *kanbun* 漢文 text may have only been comprehensible to well-educated individuals of the time (Nogueira Ramos 2020, pp. 64, 76). This readership consideration could be one of the reasons for Sessō's omission of the Confucian portions. It is possible that his fellow monks were unfamiliar with the Confucian classics, rendering it pointless or even confusing to include those parts in the original text. Additionally, during the early Tokugawa period, Neo-Confucianism emerged as a prominent political ideology in Japan, and many Confucian scholars of Sessō's time were known for their anti-Buddhist and pro-Shinto stance (Endo 2003; Kyo 1998). For example, the domains of Mito, Okayama, and Aizu all experienced anti-Buddhist movements under an ideology of Confucian-Shinto syncretism in the 1660s that saw the demolishment of many Buddhist temples. The populace in Aizu and Okayama were even ordered to register themselves with Shinto shrines instead of Buddhist temples (Ooms 1985, pp. 192–93). This could serve as another reason, assuming that Sessō intentionally omitted the Confucian sections to avoid potential criticism from Confucian scholars. Lastly, there is also the possibility that Sessō only had access to an abbreviated version of *Yuandao pixie shuo* that had already excluded the Confucian content.

Given this last possibility, it is necessary to delve into the rationale behind Sessō's selection of specific parts for inclusion in his text. To accomplish this, we must begin by scrutinizing the structure of Feiyin's original text. Feiyin's *Yuandao pixie shuo* is divided by the author himself into four sections, each bearing a title:

1.  Revealing the root of the heretical view 揭邪見根源;
2.  Revealing the heretical view that slanders the Buddha with emptiness 揭邪見以空無謗佛;
3.  Revealing the heretical view that deviates from obligation, causing confusion by dividing souls into three types 揭邪見不循本分以三魂惑世;
4.  Revealing the heretical view falsely believes that the myriad things cannot be one 揭邪見迷萬物不能為一體.

In the quoted text, parts (1) to (8) are all extracted from the first section of *Yuandao pixie shuo*. While parts (1) to (5) are presented in a continuous manner with omissions in between, parts (6) to (8) exhibit a divergence in sequencing, as Sessō places the conclusion (8) before arguments (6) and (7). Within the passage, there are missing sentences between (7) and (8), which I have enclosed in brackets. Between (8) and (6), the original text entails a logical deduction concerning the fallacy of positing a beginningless and endless God (Wu 2003).[22] Although this logical deduction is omitted, the essential meaning of the text remains unaltered. In Sessō's text, the alteration in the order of the conclusion and arguments from Feiyin's original text does not fundamentally change its significance.

Therefore, from a structural perspective, parts (1) to (8) constitute an abridged rendition of the first section of Feiyin's *Yuandao pixie shuo*.

Parts (9) and (10) exhibit a slight ambiguity when compared to the preceding sections. Part (9) comprises a quotation from the third section of the original text, while part (10) is a quotation from the second section. These two sections are notably brief and repetitive, reiterating arguments that Sessō had already presented in the text prior to his quotations.

I would like to propose a hypothesis that the entire quote primarily consists of extracts from the first section of *Yuandao pixie shuo*. Considering Sessō's limited time while interacting with Chinese monks in Nagasaki, it would have been impractical for him to transcribe the entire text. Thus, he likely chose to extract primarily from the first section, as it offers a metaphysical and cosmological perspective that supplements his existing arguments. In the preceding paragraphs, Sessō had already discussed how Christians failed to grasp the Buddhist concepts of emptiness and the unity of all phenomena, which aligns with the content found in sections (2) and (4) of *Yuandao pixie shuo*. However, Sessō did not touch upon the issue discussed in section (3), which pertains to the Christian classification of three types of souls. It is possible that Sessō overlooked this matter since, from his viewpoint, Christianity was a heretical teaching invented by Jesus, who only borrowed the names and forms of Buddhism. Consequently, Sessō may have omitted the discussion of the Christian classification of souls due to the lack of a comparable concept in Buddhism. Additionally, it should be noted that the doctrine of the three types of souls was frequently addressed in Jesuit texts both in Japan and China, often serving as a prominent point of debate between the Jesuits and local intellectuals in the early 17th century. Therefore, it seems highly improbable that Sessō was entirely unaware of this doctrine.

However, the possibility remains that someone else had previously created an extract of *Yuandao pixie shuo*, which Sessō happened to acquire. However, it would be puzzling if such an extract solely focused on the first section while randomly selecting two insignificant portions from the remaining text. Moreover, if someone else's extract were more comprehensive than Sessō's quote and Sessō extracted from this extract, it raises the question of why Sessō specifically chose part (9), which states, "Christ has lost in his original mind, detached from his original nature". This excerpt does not significantly impact the core argument in any meaningful way. Considering the aforementioned textual analysis, it appears that the most plausible explanation for the nature of the quoted sections is that Sessō extracted them directly from Feiyin's original text.

However, Sessō's modifications to Feiyin's original text indicate that his quotation was not merely a straightforward act of borrowing. Instead, it reveals a deliberate manipulation of the original text to align with Sessō's own objectives. It would be erroneous to assume that Sessō found Feiyin's anti-Christian arguments beneficial and, by quoting from him, wholeheartedly embraced Feiyin's original ideas. Conversely, Feiyin's original text was selectively omitted to suit Sessō's specific anti-Christian discourse.

## 5. Sessō's Connection with the Chinese Buddhist Community

Nishimura hypothesizes that Sessō encountered Feiyin's text during his lectures on *Rinzai roku* 臨済録 at Kōfukuji, following his preaching activities (Nishimura 2018). This particular event is documented in *Daien hōkan kokushi goroku* 大円宝鑑国師語録, a compilation of the teachings and remarks of Zen monk Gudō Tōshoku, one of Sessō's close friends. Gudō's account provides the following report:

> Sessō, stayed at Nagasaki, gave a lecture on the *Rinzai roku*. At the time, the Chinese monks listened [to this lecture], and celebrated it with poetry. I heard about the news that Sessō was helped by the monks from Ming to restore the morale of our sect. (Okuwa 1984, p. 55)

Who attended Sessō's lecture? Okuwa suggests that, during Sessō's time in Nagasaki, the Chinese monks who likely interacted with him were Wuxin Xingjue 無心性覚 (1613–1671) and Yiran Xingrong 逸然性融 (1601–1668) (Okuwa 1984, p. 312). I suggest Mozi Ruding 黙子如定 (1597–1657) should also be added to this list. Wuxin arrived in Nagasaki

and initially stayed at Kōfukuji in 1648. The following year, he relocated to another Chinese Buddhist temple in Nagasaki known as Sōfukuji 崇福寺. Wuxin hailed from Fuqing county 福清 in Fujian province 福建, the same county of origin as Feiyin Tongrong and the founder of Japanese Ōbaku Zen, Ingen Ryūki 隠元隆琦 (Ch. Yinyuan Longqi, 1592–1673). Consequently, he had a strong connection with the Chinese Huangbo branch of Chan 黄檗宗 (known as Japanese Ōbaku Zen). In fact, Wuxin played a significant role in inviting Ingen to Japan. Wu (2015) has provided a detailed study on this topic. Yiran, on the other hand, was born in Qiantang county 錢塘 in Hangzhou 杭州, Zhejiang province 浙江. He arrived in Nagasaki as a merchant in 1641 and joined Kōfukuji in 1644, where he studied under Mozi Ruding. Impressively, Yiran ascended to the position of abbot at Kōfukuji within just one year. Subsequently, he became a key figure in the invitation of Ingen to Japan (Takenuki 2020, pp. 47–48). Yiran's master, Mozi, was born in Jianchang county 建昌 in Jiangxi province 江西. He arrived in Nagasaki in 1632 and assumed the role of abbot at Kōfukuji in 1635. Although Mozi had retired by the time Sessō arrived in Nagasaki, it is possible that they had established a relationship prior to Sessō's preaching. In 1631, while Mozi was still in China, he presented a plaque as a gift to Tafukuji, where Sessō had recently become the abbot. This suggests a potential connection between Mozi and Sessō predating their interactions in Nagasaki (Okuwa 1984, p. 297).

The Nagasaki Chinese community had close connections with the Buddhist community in Zhejiang and Fujian provinces of China during that time. Kōfukuji, being established by Chinese immigrants from the Jiangnan 江南 region, which encompasses areas including Suzhou 蘇州 Nanjing 南京, and Hangzhou (Yamamoto 1983, pp. 146–47), attracted individuals such as Yiran, a merchant, and Mozi, a monk, who were based in Xingfu Chanyuan 興福禪院 in Yangzhou 揚州 near Suzhou before leaving to Japan. This geographical proximity explains their affiliation with Kōfukuji. Not only did Kōfukuji maintain strong ties with the Chinese communities from the Jiangnan area, but it also attracted monks from Fujian, such as Wuxin, due to the association with the Chinese Huangbo lineage. The Huangbo monks traced their succession back to Miyun Yuanwu, who was based at Tiantong temple 天童寺 in Zhejiang. Miyun had a significant influence in Fujian as well. In 1630, he traveled to Fujian and assumed the abbacy of Wanfu temple 萬福寺 on Huangbo mountain, where he designated Feiyin as his Dharma heir. Consequently, the students of Miyun established a robust network of practitioners in the Jiangnan and Fujian regions. Recent research conducted by Marcus Bingenheimer on Miyun's Dharma heirs reveals that their number and impact were unparalleled during the late Ming and early Qing periods, contributing significantly to the development of Chinese Buddhism (Bingenheimer 2023).

Within the interconnected network centered around Miyun Yuanwu, an anti-Christian network also emerged. Xiao Qinghe's analysis of the relationships among authors in the anti-Christian compilation *Poxieji* 破邪集 (Jp. *Hajakyū*) reveals the close association between monks and Buddhist lay believers in the regions of Zhejiang and Fujian, particularly around figures such as Yunqi Zhuhong 雲棲袾宏 (1535–1615) and Miyun (Xiao 2013). In fact, the project to compile such a collection of anti-Christian texts was initiated under the order of Miyun Yuanwu himself and his Dharma heir Feiyin Tongrong. Wu Jiang's comprehensive study on the Buddhist anti-Christian movement in late Ming China demonstrates that, while other Buddhist masters remained silent on the Jesuit mission, the Huangbo masters actively promoted anti-Christian discourses in collaboration with their Confucian literati lay believers, employing a highly organized and assertive approach (Wu 2018).

*Poxieji*, a popular anti-Christian compilation, was first printed around 1639 or 1640. It was later reprinted in 1856 by Tokugawa Nariaki 徳川斉昭 (1800–1860), the Lord of Mito. Naturally, the Buddhist community in Nagasaki maintained close ties with the compiler and many authors of this compilation. For instance, Ingen Ryūki brought an entire set of *Poxieji* to Japan in 1654 at the request of his friend and fellow student of Feiyin Tongrong, Xu Changzhi 徐昌治 (1582–1672), who was the compiler of the collection (Ito 1981). The anti-Christian treatises composed by Yunqi, Miyun, and Feiyin are all included in this

compilation. Due to the overlapping networks of the anti-Christian movement in China and the Buddhist community in Zhejiang and Fujian, it is highly likely that *Poxieji*, or some of the treaties contained within it, had already circulated within the Nagasaki Chinese community prior to Sessō's arrival.

In both of Sessō's biographies, it is reported that, after Sessō's preaching in Nagasaki, Inoue made efforts to persuade the shogun Tokugawa Iemitsu to designate Suwa Shrine 諏訪神社 in Nagasaki as the preaching location for Sessō. Unfortunately, Sessō passed away before he had the opportunity to deliver his sermons at Suwa Shrine. The biographies report that, following Sessō's death, the people of Nagasaki, who held him in high regard, sought to invite Ōbaku monks from China to take his place. This ultimately led to Ingen's arrival in Japan five years after Sessō's passing. Kengan, in both of his biographies of Sessō, considered the arrival of Ingen to be significant and added it to his accounts, suggesting the following:

> Zen Buddhism in our country was thus revived. Nagasaki's populace, when talking about the revival of the true teaching, recognize more than half of the credit to the contribution of master [Sessō]. (Okuwa 1984, p. 10)

It is important to note, as indicated by Kengan's account, that the veneration of Sessō by the people of Nagasaki was not solely based on his anti-Christian activities, but rather on his efforts to revive Zen Buddhism. Therefore, it would be inadequate to limit Sessō's legacy in Nagasaki to his anti-Christian work and the sermons he delivered. Instead, his anti-Christian endeavors should be seen as a component of his broader mission to revitalize Zen Buddhism in Nagasaki.

It is worth noting that Sessō's relationship with Chinese monks extended beyond his interactions in Nagasaki. There is evidence to suggest that his connections with monks in China may have begun well before his sermons in Nagasaki. One example of such a relationship can be seen in the association between Mozi Ruding and Tafukuji. Mozi was renowned for his calligraphy, and he was the one who inscribed the plaque at Tafukuji, where Sessō served as the abbot. The plaque bears the inscription "summer of Kanei 8 (1631)" and "monk Ruding of Xingfu Chanyuan in Great Ming" (Okuwa 1984, p. 297). According to *Tafukuji gyōyū*, a biographical record of the first three abbots of Tafukuji preserved at the temple, Sessō assumed the position of the second abbot of Tafukuji in the spring of Kanei 8 and commenced the reconstruction of the temple (Okuwa 1984, p. 20). The plaque from Mozi, therefore, likely served as part of the efforts to rebuild the temple. It is worth noting that Mozi himself only arrived in Nagasaki in 1632, indicating that the Xingfu Chanyuan mentioned on the plaque was the temple where Mozi resided in China at that time. Consequently, it can be inferred that Sessō had already established connections with monks in China prior to 1631.

Furthermore, Okuwa's research highlights the close association between Sessō's understanding of the unity of Zen, Pure Land, and Ritsu 律 schools of Buddhism and the practice of *nenbutsu zen* 念仏禅 advocated by the Ōbaku school (Okuwa 1984, p. 312). Given their shared perspective on Buddhism, it is only natural that Sessō would establish friendships with the Chinese monks in Nagasaki and engage in the study of texts produced by Miyun and Feiyin, who were Ingen's two masters. Additionally, from an institutional standpoint, the Myōshinji branch, to which Sessō belonged, maintained a strong connection with Miyun and his disciples. In fact, when Ingen arrived in Japan, there were even discussions within the Myōshinji branch about appointing him as the abbot of Myōshinji (Nogawa 2016, pp. 234–50). Therefore, it comes as no surprise that Sessō had ample opportunities to establish connections within the Chinese Buddhist community in Nagasaki. This further elucidates why we find many overlapping arguments in Sessō's writings and the anti-Christian treaties of Chinese monks. It also suggests that Sessō likely had the chance to read Feiyin's *Yuandao pixie shuo*, as well as other Chinese Buddhist anti-Christian texts, in their entirety.

Thus, contrary to a mere adoption of ideas from Chinese anti-Christian treaties, the relationship between Sessō and the Chinese monks can be argued to be rooted in their

shared goal of revitalizing Zen Buddhism. For instance, according to Gudō's account, Sessō played a crucial role in restoring the morale of Rinzai Zen by delivering lectures on *Rinzai roku* in Kōfukuji. This endeavor was clearly aimed at the advancement and fortification of Buddhism, rather than having a singular focus of dismantling Christianity. Furthermore, Sessō's accomplishment in Nagasaki involved facilitating tens of thousands of the local populace in receiving the Buddhist Five Precepts, an act that can be likened to baptism in Christianity. The significance lies not in the conversion of hidden Christians, but rather in the number of individuals who chose to embark on their Buddhist journey and embrace the path of lay believers.

When examining the early Tokugawa anti-Christian persecution, it is frequently contextualized within a socio-political framework that underscores trade dynamics, national stability, and apprehension over foreign intrusion. This perspective positions Buddhism as an instrument that the shogunate harnessed to exert societal control and oversight, as elucidated by Paramore (2009) and Hur (2007). This top-down vantage point interprets the early Tokugawa anti-Christian crackdown as a pivotal strategy employed by the shogunate to consolidate its political dominance.

However, an assessment of the individuals responsible for executing this anti-Christian mandate reveals that their motivations often extended beyond mere political considerations. Take Sessō, for instance; he nurtured his aspiration to reinvigorate Zen Buddhism in Japan. This lifelong pursuit manifested not only through his affiliation with a group of like-minded revolutionary monks in Japan, but also through his close involvement with or ties to the resurgence of Chinese Chan Buddhism. Consequently, Sessō's anti-Christian compositions encompass a distinct dimension beyond the official political concerns: the shared objective of Zen/Chan revitalization enabled Sessō to engage with Chinese monks and, through this engagement, access Chinese anti-Christian writings that he incorporated into his own mission.

As a result, an alternate perspective from the bottom–up can complement the conventional top–down viewpoint. While Buddhism has frequently been portrayed as a tool for political control within the early Tokugawa anti-Christian crackdown, the personal religious inclinations of Buddhist monks tasked with anti-Christian endeavors could intersect with their official responsibilities, thus mutually benefiting both realms.

## 6. Other Potential Influences from the Chinese Anti-Christian Works in *Taiji jashū ron*

Other than the quotation from Feiyin's text, additional elements in *Taiji jashū ron* bear resemblance to those found in Chinese Buddhist anti-Christian texts. For instance, Sessō states in *Taiji jashū ron* that the early Japanese brother Lorenzo[23] "departed from Sasshū (薩州) to Rome to learn *tenshugyō* (天主教) and came back to Japan. This sect is called Kirishitan (喜利志祖). Lorenzo represented the padres to teach *tenshugyō* and attract people to convert to this sect (宗門), the number of believers amount to more than one hundred" (Okuwa 1984, p. 88). This description is notable due to Sessō's usage of both *tenshugyō* and Kirishitan to refer to Christianity, since the term *tenshugyō* was rarely employed, if at all, in Japan during that period.

According to Ebisawa Arimichi (1910–1992), the Japanese Jesuits initially employed certain Confucian terms to denote God when they realized that the term they initially used, Dainichi 大日,[24] was misleading. These alternative terms, which included *tendō* 天道, *tentei* 天帝, and *tenshū* 天主, were used from the 1560s to the early 1590s to replace Dainichi. However, in 1592, with the publication of the catechism *Dochirina Kirishitan*, the Latin word for God, Deus, was deemed the correct term to refer to God in Japan (Ebisawa 1971, pp. 269–71). On the other hand, the term *tianzhujiao* (equivalent to *tenshugyō*) gained popularity through Ricci's influence in China. It was vehemently criticized by the Jesuit fathers who were compelled to leave Japan for Macao and China in the 1610s, following Tokugawa Ieyasu's prohibition of Christianity (Cooper 1974, pp. 277–83; Farge 2012). A study of other anti-Christian texts produced in the first half of the 17th century also reveals that they referred to the Christian religion as Kirishitan, rather than *tenshugyō*. Conversely, due

to Ricci's influence, *tianzhujiao* became widely used in China by both Buddhist monks and other individuals. This term is commonly found in the treatises of Yunqi, Miyun, Feiyin, and other monks' anti-Christian writings. Therefore, Sessō's utilization of this term suggests his familiarity with Chinese anti-Christian texts.

Yunqi Zhuhong's influence on Sessō's understanding of the Christian God should also be considered. For instance, in the early 1610s, Yunqi identified the Christian God as a Buddhist deva, specifically, Tāvatimṣa, in his anti-Christian work *Tianshuo* 天說.[25] In *Taiji jashū ron*, Sessō expresses a similar notion that God is merely a Buddhist deva:

> [Jesus] stole dharma from the Buddhists and adapted it to the heretical teaching. He either changed its name but held onto its essence or maintained its form but altered its principles. So he changed the name of Brahmā (梵天王) to Deus (泥烏須), the heavenly deities (梵衆) to angels (安助), heaven to paraíso (頗羅夷曽), earthly world to Purgatório (跀嬰伽倒利夜), hell to inferno (因辺嬰濃)… (Okuwa 1984, p. 92)

Although Yunqi Zhuhong and Sessō identified God as different Buddhist devas, their underlying idea is the same: they both argue that the Christian teaching mistakenly attributes the origin of the cosmos and all phenomena to a single deity without deeper understanding or consideration.

Furthermore, Sessō regarded Christianity as a false teaching that employs a specific set of doctrines to impose restrictions on individuals (与一定法，縛住諸人). He criticized Christianity for disregarding the enduring nature of Suchness (真如常住) and the perpetual consequences of karma (因果不亡), while promoting the existence of a Heavenly Lord (天主) and the notion of an afterlife (後世). In his conclusion, Sessō argued that heavenly rewards and other forms of retribution are the direct outcome of human karma (蓋天堂等報，因人造業而有). Miyun Yuanwu notably presented a similar argument in his *Biantian shuo*, asserting that Heaven and Hell are the result of human karma (夫天堂地獄，蓋衆生業力所召). According to Miyun, Buddha prescribed the Threefold Training (戒定慧之教) precisely because he comprehended the origin of Heaven and Hell (Zhong and Xu 1984, p. 11537). Sessō shared this understanding and posited that the Three Poisons[26] (三毒) in the human mind give rise to ten forms of negative karma (十悪業), leading individuals to traverse the Three Evil Paths (三悪道). The Threefold Training in Buddhism serves as a transformative practice aimed at addressing and transcending these Three Poisons, demonstrating the convergence of Sessō and Miyun's perspectives.

## 7. Conclusions

The apprehension regarding the influence of European culture and Christianity upon Japan had consistently remained a paramount concern for the Tokugawa shogunate. Throughout the duration of the Tokugawa era, the government fervently pursued the objective of uncovering and eradicating any residual Christian presence within the nation. In the preface to the 1856 edition of *Poxieji* by Tokugawa Nariaki, he expressed his concern over this matter:

> When the evil teaching [Christianity] entered the Holy Nation [Japan] in the past, Hideyoshi tried to ban it and Ieyasu tried to wipe it out of Japan. Hidetada and Iemitsu both filially succeeded their ancestors' will and enterprise. However, the Shimabara Rebellion happened precisely during the most peaceful time … After the rebellion, the prohibition became even more strict. Whenever came an evil person [missionary], that person was killed, and his boat burnt. Therefore, the shogunate had destroyed the barbarians' hope of converting Japan. Compare to the Late Ming in the west where the missionaries were only expelled rather than annihilated and kept out, the shogunate's achievement was much greater. (Kanzaki 1893, p. 133)

After the passage of over two centuries, the Lord of Mito would employ the contents of *Poxieji*, much like Sessō's utilization in Nagasaki. This serves as a testament to the en-

during significance attributed to Chinese anti-Christian discourse within Japan. Nariaki commended the resolute measures undertaken by the shogunate while expressing regret over the lenient stance adopted by the Ming government. Undoubtedly, Sessō's sermons formed an integral component in the shogunate's decisive strategy to implement the *shū-mon'aratame* system in Nagasaki.

Sessō effectively fulfilled his mission by leveraging the anti-Christian literature available in Japan and China during that period. Although indigenous influences have been subject to scholarly scrutiny for decades, the Chinese influence on Sessō's anti-Christian ideology has only recently received attention. Nishimura, as the pioneering scholar shedding light on Sessō's quotation of Feiyin, focused her research on examining the intellectual impact discernible within the quotation. She identified Feiyin Tongrong as an individual who exerted influence upon Sessō. While Sessō's text does indeed demonstrate a direct influence from Feiyin, it is imperative to recognize the existence of a broader network of individuals involved in the Buddhist anti-Christian movement, with Feiyin assuming a significant role within it. Hence, beyond Feiyin's direct influence, it is conceivable that Sessō encountered a wider discourse of anti-Christian sentiment in China.

This study draws upon Nishimura's discoveries and centers its investigation on the evolution of Chinese anti-Christian discourse. During his time in Nagasaki, Sessō engaged with Chinese monks who maintained robust ties with the Huangbo monks residing in China. This connection provided him with access to texts generated within the anti-Christian movement spearheaded by the aforementioned Huangbo monks. By considering this interpersonal network, we can now attain a more comprehensive comprehension of the transmission of anti-Christian discourse from China to Japan during the early Tokugawa period.

In conclusion, this article establishes three key points. Firstly, the inclusion of Sessō's quote from Feiyin does not imply a wholesale acceptance of Feiyin's ideas, such as the syncretism of Confucianism and Buddhism. Sessō modified Feiyin's text to align it with the distinctive intellectual and socio-political context prevailing in Japan during his era. Secondly, Sessō's quotation of Feiyin served as a means to propagate not merely the anti-Christian views of an individual Chinese monk, but rather a broader discourse embraced by a closely interconnected group of devout Buddhist adherents in China. Thirdly, Sessō's interaction with Chinese monks in Nagasaki facilitated his engagement with a network of Chan monks in China, with a particular focus on Miyun Yuanwu. While their exchange encompassed various aspects of Buddhism, their collaboration extended beyond anti-Christian endeavors, reflecting a collective endeavor to revitalize Zen/Chan Buddhism.

**Funding:** This research received no external funding.

**Institutional Review Board Statement:** Not applicable.

**Informed Consent Statement:** Not applicable.

**Data Availability Statement:** Not applicable.

**Conflicts of Interest:** The author declares no conflict of interest.

## Appendix A

For Sessō's part, see Okuwa, *Shiryō kenkyū sessō Sōsai*, pp. 101–3. For Feiyin's part, see Zhong and Xu, *Kinsei Kanseki Sōkan: Wakoku Eiin Shisō 4-hen 14*, pp. 11573–98. The underlined part highlights the directly quoted parts from Feiyin's text by Sessō.

| 対治邪執論 | 原道辟邪说 | Translation |
| --- | --- | --- |
| (1) 三身寿量無边経曰、妙覚毘盧遮那、承無始無終一心一念本仏説法、汝曰、天主則無始無終、而為万物始焉、殊不知、＜便是喜＞利志徒妄執無始無終、為天主之邪見根本矣、殊不知、此無始無終正是吾大道之元、亦是吾全真之旨、且此全真之旨、人々具足、大道之元、個々不無、在聖無増、処凡不減、抑亦在天而天、在人而人、至于物々如是、法々亦然、固無二無二分、無別無断、故悟此謂之聖人、迷此謂之凡夫、要且凡夫之与聖人、初無二致、如是則聖凡靡間、而物我匪虧、顕見大道之元、無彼無此、全真之体、無始無終、一道平等、而浩然大均矣、喜利志徒、不悟此意、専用心意識、向天地万物上、推窮計度、到虚玄深貌処、自家躰貼不来、便妄執有箇天主、具無始無終之量、能育天地、健生万物、而万物則有始有終、謂鳥獣草木是也、有始無終、則天地鬼神及人之靈魂是也、惟天主無始無終、能制造万物、 | 按利瑪竇邪見，妄著《天主實義》一書，列為八篇。而首篇論天地萬物布置安排，皆由天主所生。論至天主則曰，天主之稱，謂物之原，如謂有所生則非天主也。物之有始有終者，鳥獸草木是也；有始無終者，天地鬼神及人之靈魂是也；天主則無始無終，而為萬物始焉。據此便是利瑪竇妄執無始無終為天主之邪見根源矣。殊不知此無始無終，正是吾大道之元，亦是吾全真之旨。且此全真之旨，人人具足；大道之元，個個不無。在聖無増，處凡不減；抑亦在天而天，在人而人。至於物物如是，法法亦然，固無二。無二分，無別無斷。故悟此謂之聖人，迷之謂之凡夫。要且凡夫之與聖人，初無二致。如此則聖凡靡間，而物我匪虧，顯見大道之元無彼無此，全真之體無始無終，一道平等而浩然大均矣。蓋瑪竇不悟此意，專用心意識向天地萬物上妄自推窮計度。以心意識向天地萬物上推窮計度到虛玄深貌處，自家體貼不來，便妄執有個天主具無始無終之量，能育天地，健生萬物。而萬物則有始有終，謂鳥獸草木是也，有始無終則天地鬼神及人之靈魂是也。惟天主無始無終，能制造斡旋。 | The Sutra of the Immeasurable Lifespan of the Three Bodies says, "In the sublime enlightenment of Vairocana, based on the Buddhist doctrine of beginningless and endless of the one heart and one thought." You say, God is beginningless and endless, and is the origin of all things. But you don't know, this insistence on a beginningless and endless God is a fundamental heresy held by Christ. Unbeknownst to you, this beginninglessness and endlessness is indeed the origin of the Great Way, and also the purpose of the ultimate truth. Moreover, this ultimate truth is within everyone, and the origin of the Great Way is not absent in anyone. It does not increase in the saints, nor decrease in the ordinary people. It exists in the sky as the sky, in humans as humans. As for all things, so it is, and for all phenomena, it is also the same. Thus, there is no duality, no division, no distinction. Therefore, those who understand this are called saints, and those who are confused by this are called ordinary people. Furthermore, there is no essential difference between ordinary people and saints. In this way, there is no division between the saints and the ordinary people, and neither things nor self are deficient. The origin of the Great Way is clearly visible, without this or that. The body of the ultimate truth is beginningless and endless, the Way is equal, and, hence, a great harmony is formed. Christ does not understand this concept and uses his mind and consciousness to explore and estimate everything within heaven and earth. When he reaches the vast and mysterious depths, he cannot quite grasp it. Therefore, he arbitrarily assumes the existence of a God, an entity with no beginning and no end, capable of giving birth to heaven and earth, and fostering all beings. Further, [he teaches] all these beings indeed have a beginning and an end, such as birds, animals, plants, and trees. And there are entities that have a beginning but no end, such as the earth, heaven, ghosts, gods, and human souls. Only God has no beginning and no end, capable of creating everything. |

| 対治邪執論 | 原道辟邪説 | Translation |
|---|---|---|
| (2) 窃仏説無始無終之語、以虚妄之心、推窮計度、而引誘多方、党于邪見、 | （且指物比類，要人欽奉遵守。而矯為過高之論，卑劣今古聖賢。指人都無有主，）而引誘多方，黨於邪見，（假詞击难，辨驳繁端，不啻枝上生枝而蔓上生蔓。） | [Christ] stealthily borrows the language of Buddhism that speaks of no beginning and no end, using a deceitful heart to speculate and measure, thereby luring many and leading them to follow erroneous views. |
| (3) 仏説無始無終者、因人契証、以顕人物天地及其鬼神俱是無始無終底意耳、就当人心念上、返照窮元、則過去心念無有、而未来心念無起、現在心念無住、三際既無、則心念全無始、而亦全無終矣、如心念既無始、而又無終、則身体脱然無繋、亦無前後三際、了無生死去来、直下披露無始無終、即色身五蘊完全解脱、而大道全真備在我＜矣＞、既人々返照窮元、契無始終、則草木鳥獸天地鬼神、当前廓爾、貌無形跡、便是草木等類、全無始終、而顕大同之旨也、 | （或云：「人物鳥獸與天地鬼神，如何見得是無始無終之旨耶？」曰：「前已總明，今又複問，姑分二說：一者，）因人契証，以顯人物天地及其鬼神，俱是無始無終底意耳。就當人心念上返照窮元，則過去心念無有，而未來心念無起，現在心念無住，三際既無，則心念全無始而亦全無終矣。若心念既無始而又無終，則身體脱然無系，亦無前後三際，了無生死去來。直下披露無始無終，即色身五蘊，完全解脱；而大道全真，備在我矣。既人人返照窮元，契無始終，則草木鳥獸天地鬼神，當前廓爾，貌無形跡，便是草木等類全無始終，而顯大同之旨也。 | When the Buddha speaks of "no beginning and no end," it is due to humans' personal experience and enlightenment, showing that human beings, all things, heaven, earth, and their spirits are all without a beginning and an end. Reflecting upon one's thoughts, tracing back to the origin, there is no past thought, no future thought arises, and no current thought resides. With these three realms of thought being nonexistent, thoughts are entirely without a beginning and equally without an end. If thoughts have neither a beginning nor an end, the physical body is completely unbound, also without past, present, and future realms. It has neither birth nor death, coming nor going, revealing directly the concept of "no beginning and no end." Thus, the physical body and the Five Aggregates are completely liberated, and the complete truth of the Great Way exists within me. If every person reflects upon the origin and understands the idea of "no beginning and no end," then plants, animals, heaven, earth, ghosts, and gods, all broads and forms in front of us, are, in fact, without form or trace. This means that entities like plants have no beginning and no end, revealing the purpose of the Great Harmony. |

| 对治邪執論 | 原道辟邪说 | Translation |
|---|---|---|
| (4) 鬼神天地鳥獸草木、雖因人契証顕其無始無終、要且自性如是、而亦自離意言境、故経曰、諸法不自生、亦不従他生、不共不無因、是故説無生、生生之体、渾然一致、黙識心通、而与契合、無容妄想執著擬議分別于其間矣、又明天地人物及其鬼神、不因人証、本来是無始無終、全無間隔之差、且拠実約多広而論、則虚空無尽、而所包世界亦無尽、以所居衆生亦無尽、乃至天地鬼神草木鳥獸、悉皆無尽、不得而数量之、以虚空無有辺際、則凡所有物、悉無辺際、法爾如是、又拠実約久常而論、則虚空無終始、而世界亦無終始、衆生亦無終始、天地鬼神草木鳥獸、悉無終始、覓其終始起伏、了不可得、以顕虚空世界一切衆生、及天地鬼神草木鳥獸、同時同際無分前後、永久常存、熾生不息、蓋亦不期然而然、 | （又「天何言哉，四時行焉，百物生焉。」則天亦無識心分別，故能行四時生百物，而與四時百物冥相溥洽，更無缺悖者矣。又「鬼神之為德，其盛矣乎！視之而弗見，聽之而弗聞，體物而不可遺。」夫鬼神既非視聽可及，又能體物不遺，則鬼神亦無識心分別，而其德固為盛也。且孔子推鬼神之德如此之盛，而瑪竇謂有始無終，豈其宜乎？然則）鬼神、天地、鳥獸、草木，雖因人契証顯其無始無終，要且自性如是，而亦自離意言境。故經云：「諸法不自生，亦不従他生。不共不無因，是故說無生。」無生之體渾然一致，黙識心通而與契合，無容妄想執著擬議分別於其間矣。（二者，）以明天地、人物及其鬼神不因人証，本來是無始無終，全無間隔之差。且據實約多廣而論，則虚空無盡，而所包世界亦無盡，以所居眾生亦無盡，乃至天地、鬼神、草木、鳥獸悉皆無盡，不得而數量之。以虚空無有邊際，則凡所有物悉無邊際，法爾如是，（非是強為使之然也）。又據實約久常而論，則虚空無終始，而世界亦無終始，眾生亦無終始，並及天地、鬼神、草木、鳥獸悉無終始。覓其終始起伏，了不可得，以顯虚空世界一切眾生及天地、鬼神、草木、鳥獸，同時同際，無分前後，永久常存，熾生不息。蓋亦不期然而然，（非使之然也）。 | Spirits, gods, heaven, earth, birds, animals, and plants, although their lack of a beginning and an end is revealed through human testimony, it is also inherent in their nature and beyond the realm of conscious, thought, and speech. Therefore, the sutra says, "All phenomena do not arise by themselves, nor are they born of others. They are neither conjoint nor without cause. Therefore, it is said they are unborn." The essence of all life is united, in sync with our silent awareness, leaving no room for arbitrary attachments or discriminations within. Furthermore, it clarifies that heaven, earth, all beings, and their spirits do not rely on human testimony; they are intrinsically without a beginning and an end, with absolutely no distinction in between. Moreover, if we discuss based on a wide and expansive reality, the cosmos is infinite, and so are the worlds it encompasses, and the beings that inhabit these worlds. Even heaven, earth, spirits, gods, plants, birds, and animals, all are infinite, beyond any quantification. Since the cosmos has no boundaries, all things within it also have no boundaries. Thus, this is for all phenomena. And speaking in terms of duration, the cosmos has no end or beginning, and nor do the worlds, the beings, the heaven, earth, spirits, gods, plants, birds, and animals, all have no end or beginning. Any attempt to find their beginnings, endings, rises, or falls will ultimately be fruitless. This reveals that the cosmos, all the worlds, all beings, along with heaven, earth, spirits, gods, plants, birds, and animals, exist simultaneously without differentiation of before and after, enduring eternally, continuously creating life, just as it happens, not expected but inevitably so. |

| 対治邪執論 | 原道辟邪説 | Translation |
|---|---|---|
| (5) 心也者、総持之大本、万法之洪源、不可以智知、不可以識識、智莫能知、識莫能識，默契其旨、存乎其人也。 | （故孟子曰：「萬物皆備於我」，子思曰：「致中和，天地位焉，萬物育焉。」程子曰:「放之則彌六合，卷之則退藏於密。」經亦曰:「心生則種種法生，心滅則種種法滅。」又云：「）心也者，總持之大本，萬法之洪源，不可以知知，不可以識識。知莫能知，識莫能識，默契其旨，存乎其人也。」 | The mind, in this context, is the great origin that upholds everything, the vast source of all phenomena. It cannot be understood by wisdom, nor can it be perceived by consciousness. Wisdom cannot know it, and consciousness cannot perceive it. Its purpose is silently realized and lies within each individual. |
| (6) 喜利志徒、全不省天地万物備于自己、而自己与天地万物定無始無終本来者一著子、向天地万物之外、妄執有一天主独具無始無終、誠為邪見外道也、 | 然瑪竇全不省天地萬物備於自己，而自己與天地萬物具足無始無終本來者一著子，向天地萬物之外，妄執有一天主獨具無始無終，誠為邪見外道也。 | Christ entirely fails to realize that heaven, earth, and all things exist within oneself, and that the self and heaven, earth, and all things inherently share the nature of being without a beginning and an end. Instead, he arbitrarily clings to the belief that there is a single God distinct from heaven, earth, and all things, solely possessing the characteristic of no beginning and no end. This is indeed a heterodox and erroneous view. |
| (7) 蓋万物既有最初始生之時、則最初始生之前、無有万物、既無有万物、則必彼時天主能生之功、亦必有滅有終、以因天主能生之功、有滅有終故、顕万物最初始生之前無有、既彼時無有能生之功、又無所生之物、則顕無有天主、唯一混沌空晦而已、衆生召感、混沌空劫是也、 | （且瑪竇妄執有天主獨具無始無終，而生萬物為有始有終，理甚乖舛，誠不足信，試以辨明。）蓋萬物既有最初始生之時，則最初始生之前，無有萬物。既無有萬物，則必彼時天主能生之功，亦必有滅有終，以因天主能生之功有滅有終，故顯萬物最初始生之前無有。既彼時無有能生之功，又無所生之物，則顯無有天主，唯一混沌空晦而已。（照如前論），眾生召感混沌空劫是也。（而瑪竇不悟，錯認妄計為天主以具無始無終，寧不邪謬之甚乎？） | Since if all things indeed have a time of initial birth, then before this initial birth, there was no existence of these things. As there were no myriad things, the capacity of God to give birth to them must have also had an end and extinction. Because of the end and extinction of God's capacity to give birth, it is clear that there was no existence before the very first birth of all things. Since there was no ability to create at that time, and there was nothing that is created, it is evident that there was no God, only a primordial chaos and darkness. This is the cosmic epoch of chaos that all beings recall and respond to. |
| (8) 若有間隔空缺于其中、則非是健生不息之道、亦非全智全能之理、而亦瘉顕非具無始無終之体量也、譬如虛空該羅万象、無<時>間離、而亦無可逃遁、直与万象無始無終、方称全功、豈有天主具無始無終、為全智全能、而独生物有間有欠、有減有終乎、 | （且伊既謂天主具無始無終，則應智能體用悉無始終，方顯為全智全能，有健生不息之道。）若有間隔空缺於其中，則非是健生不息之道，亦非全智全能之理，而亦愈顯非具無始無終之體量也。譬如虛空該羅萬象，無時間離，而亦無可逃遁，直與萬象無始無終，方稱全功。豈有天主具無始無終為全智全能，而獨生物有間、有缺、有減、有終乎？ | If there were gaps or emptiness within [the phase of chaos and the phase of phenomena], then this would not be the path of continuous creation, nor the principle of omniscience and omnipotence, and it would further highlight the lack of a characteristic of being without beginning and without end. For instance, the cosmos and all its myriad phenomena are inseparable from time and have no place to escape. Only when they are coextensive with the beginninglessness and endlessness of all phenomena can they be considered complete in their function. How can there be a God that is without beginning and without end, omniscient and omnipotent, and yet the beings he creates have gaps, deficiencies, diminutions, and endings? |

| 对治邪執論 | 原道辟邪说 | Translation |
|---|---|---|
| (9) 喜利志徒、迷于本心、失于本性、 | （然則瑪竇）迷於本心，失於本性，（理必悖嘗逆倫。致君為愚，使臣不忠，而上下不和，凡天下之事悉皆倒置，必自利瑪竇輩。向外多事不循本分之故也。） | Christ has lost his original mind, is detached from his original nature. |
| (10) 作無主孤魂、計心外有一天主百年之後往彼依附、使一切人都作無主孤魂、悉如汝者、真所謂業識茫々、無本可拠也。 | （汝若是誠是有是真實，決不自甘）作此無主孤魂，計心外有一天主，百年之後往彼依附。使一切人都作無主孤魂，悉如汝者，真所謂業識茫茫無本可據也。 | [Christ] becomes an ownerless wandering soul. He imagines there is a God outside his mind that he can rely on after death. Causing all people to become ownerless wandering souls, just like him. This is truly what is meant by the ignorance of karmic consciousness, having no foundation to rely upon. |

## Notes

1. Ieyasu offered 350 taels of gold and subsequently made a loan of 5000 taels to help the Jesuits with their financial difficulties.

2. The precise count of fatalities remains a topic of ongoing scholarly debate. The conventional estimate places the figure at 37,000, while some researchers, including Gonoi, align with Nakamura's assessment of approximately 23,900 casualties.

3. Okuwa meticulously investigated the chronological sequence of Sessō's sermons in Nagasaki, proposing that he initiated the initial sermon on the date corresponding to Shōhō 4.5.6 and concluded the final one on Shōhō 4.5.28. This timeframe corresponds to the period spanning from 8 June 1647 to 30 June 1647.

4. George Elison also briefly discussed Sessō's *Taiji jashū ron* in his 1973 work *Deus Destroyed*, but his focus in that part was on another anti-Christian monk, Suzuki Shōsan 鈴木正三 (1579–1655).

5. Okuwa's *Shiryō kenkyū sessō Sōsai* includes the two anti-Christian texts of Sessō, three documents in Tafukuji related to Christianity, Sessō's biography, Sessō's other writings, and several commentaries on the biography and thoughts of Sessō.

6. The following introduction to Sessō's life is mainly based on two biographies that were penned by his successor to the abbacy of Tafukuji, Kengan Zenetsu 賢巌禅悦 (1618–1699). One is titled *Sessō oshō gyōjō* 雪窓和尚行状 and the other is titled *Butchi hisho zenji gyōjō* 仏智丕昭禅師行状. Both texts are available in Okuwa, *Shiryō kenkyū sessō Sōsai*, pp. 5–17.

7. The practice of nenbutsu involves the repetitive recitation of the phrase *namu amida butsu* 南無阿弥陀仏, with the aim of achieving rebirth in the pure land through the benevolent intervention of the Amitābha Buddha.

8. Tafukuji was once left unattended with the death of its first abbot Ryōshitsu Shūmitsu 了室宗密 (d. 1616), who was the master of both Sessō and Inaba Kazumichi.

9. The content enclosed within brackets directly originates from the original text of *Taiji jashū ron*. In the absence of any explicit markings, all text presented in this manner signifies verbatim excerpts from the *Taiji jashū ron* source.

10. For the early stages of the Jesuit mission, see Gonoi, *Nihon kirisutokyō-shi*, pp. 35–112, or Boxer, *The Christian Century in Japan*, pp. 1–90.

11. Texts in brackets is from *Yuandao pixie shuo*.

12. The full text of *Tianxue chupi* can be found in Xu, *Kinsei Kanseki Sōkan: Wakoku Eiin Shisō 4-hen 14*, pp. 11619–50.

13. A translation and comment on *Pixie ji* is available in Jones, "*Pì xiè jí* 闢邪集: Collected Refutations of Heterodoxy by Ouyi Zhixu (蕅益智旭, 1599–1655)".

14. The record of this debate is titled *Hayaso* 排耶蘇 by Razan. The full text is available in Ebisawa, *Nihon shisō taikei 25: Kirishitan-sho, haiyasho*, pp. 414–17.

15. See Itō, "Kinsho no kenkyū" Part 1, p. 13.

16. I have enclosed certain omitted sections within brackets; however, note that not all of these bracketed portions constitute complete segments that were omitted.

17. See the translation in Appendix A. Later quotations of Feiyin, if not marked, can be found in Appendix A.

18. The first quote is from the *Analects* 論語 and the second quote is from the *Doctrine of the Mean* 中庸. For the books mentioned, see Chen Junying ed. 2010. *Sishu wujing yizhu* 四书五经译注.

19. Wu Jiang suggests that the Buddhist–Christian rivalry in Late Ming China was mainly placed on the problem of orthodoxy 正, which, in that period, was unquestionably represented by Confucianism in China. He argues that both Buddhist monks and the missionaries tried to align themselves with the Confucian scholars in order to claim the orthodoxy of their own religions.

[20]    The first quote is from *Mengzi* 孟子, the second is from *Doctrine of the Mean*, and the third is from *Zhongyong zhangju*, a commentary on *The Doctrine of the Mean* by Cheng Yi 程颐 (1033–1107). The sutra quoted here is from *Awakening of Faith in the Mahayana* 大乘起信論. For the books mentioned, see (Chen 2010; Chen and Chen 2000; Yoshihide 2014).

[21]    In the previous text, Sessō insisted that Jesus failed to understand the truth of Buddhism, but sought a Creator that existed outside of this world. He based his refutation from a Buddhist viewpoint and suggested that the three realms of existence are only mind, and the myriad things are only consciousness (三界唯心，万法唯識). The three realms correspond to the three poisons of humanity (三界者即三毒也): confusion 癡, greed 貪, and aversion 瞋. Thus, the existence of the three realms is the mental projections of negative human emotions, and Jesus did not understand this, so he created a vain deception (虛妄巧見).

[22]    Due to spatial constraints, it is not feasible to offer an exhaustive analysis of this specific logical deduction within the Buddhist tradition. Those intrigued by this subject matter can access a comprehensive introduction and analysis of the said text in Wu's article.

[23]    In Sessō's text, the name is 路連曾了西, probably referring to ロレンソ了斎 (1526–1592), one of the most influential Japanese brothers in the Jesuit mission in Japan.

[24]    Dainichi refers to the Buddha Mahāvairocana 大日如来.

[25]    The full text of *Tianshuo* is available in Xu, *Kinsei Kanseki Sōkan: Wakoku Eiin Shisō 4-hen 14*, pp. 11491–98.

[26]    The Three Poisons in Buddhism refers to delusion, greed, and hate.

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
