# Peer review of "Sessō Sōsai and the Chinese Anti-Christian Discourse"

_religions, doi:10.3390/rel14081058_

Round 1
Reviewer 1 Report
I finished reading this article and enjoyed it very much. I found the article well-researched, nicely written, concise, and strongly argued.
In particular, section 5 and section 6 provide insightful details on the transmission of the Chinese anti-Catholic discourse between coastal China and Nagasaki along the religious and merchant networks.
In addition, the sources listed in the Appendix are immensely helpful for researchers, and I thank the author for sharing the documentary evidence.
As the author makes good use of the story of Sessō’s time in Nagasaki to explain the profound influence of the circulated Chinese anti-Catholic texts in Japan. It is worth mentioning that Antoni J. Ucerler, Soh Jeanhyoung, and others also document similar anti-Catholic and Catholic theological book networks that spread from China and the Philippines to Korea and Nagasaki around the same period.
In short, this manuscript is top quality. I don’t think the author need any major revisions.
Author Response
Dear Reviewer,
I would like to express my appreciation to your kind review and your precious time spent on my paper.
Thank you for your kind words.
Reviewer 2 Report
This is a very interesting and well-written article. But key issues of argumentation, organization, and historiography (or situation within existing scholarship) must be resolved before its contributions will be clear and it will be publishable.
The research question is not entirely clear, although it becomes clearer and clearer as the paper goes on. In the Introduction, the author should draw from the latter paragraphs and conclusion to state the research question clearly.
Historiography/Literature Review: In connection with the research question, the following questions should be answered in the introduction as well: Why is research needed on Sessō? What questions have been left unanswered by previous scholarship? This will require a clearer discussion of existing scholarship as well. Some scholars are discussed, but the relationship of that scholarship to the author's work is murky at best, and inconsistently described. Nishimura, Nogeira Ramos, and Okuwa are discussed at several points, but only towards the end does the author explain that they are building on Nishimura's work. Why does Nishimura's work need to be built upon? What remaining questions does her work leave? For the works mentioned, more careful analytical discussion of "what's left to learn about Sessō" and how this study can contribute to that is needed in the early paragraphs of the article.
Some key scholars are also missing, particularly with regards to Tokugawa anti-Christian thought and policy: they include Kiri Paramore's Ideology and Christianity in Japan (2009), Nam-Lin Hur's Death and Social Order in Tokugawa Japan, chapters from John Been and Mark Williams's edited volume Japan and Christianity: Impacts and Responses (esp. Kaiser and Ōhashi).
Thesis statement: There are several fragments, particularly towards the end of the article, that could be clarified and combined into a thesis. I recommend that the author state the thesis clearly towards the end of the introduction and that the thesis be made to clearly respond to the research question and the current lack in the existing research (esp. by Nishimura)
The paper should also try to group together key basic historical contextual information: Who was Sessō? What was he doing before he went to Nagasaki? Did his anti-Christian activities/discourse precede this? Who sent him? (we only read that he was sent by "the shogun" on p. 6 of the article) What was his relationship to/connection to anti-Christian ideas and persecution in Japan? Did the shogunate send other Buddhist priests to preach/write against Christianity in Kyushu? What was Sessō's connection with Inoue Masashige? Who was his audience (intended and perhaps broader)? And why was he important? (again we learn that he had thousands of listeners, and that he wanted to revitalize Zen in Japan, but very late in the paper). This kind of info will help the reader understand why he was important and why a study on him at all is important (in addition to the author's need to explain why another study on Sessō is important, as mentioned above).
In addition, at least a few sentences are needed to explain the origins of anti-Christian rhetoric amongst Zen priests in both China and Japan. A brief mention is made regarding Japan, and that could be built upon, but there is no mention of why and when Chinese T'ian priests/monks preached and wrote against the religion.
I recommend that the author devote paragraphs to Sessō's relationships with Chinese monks/priests and perhaps Japanese ones too before making the argument that these networks influenced his texts. The same also needs to be done for Nagasaki-Jiangnan/Fujian Buddhist temple and cleric's networks (with references). This info is some of the most interesting for the reader and some of the most important in order for the author to convincingly demonstrate their argument(s). So the reader needs this info before.
Throughout, the author should try to make dates and chronology clearer. At one point the author jumps to anti-Buddhist measures in the "1660s," which is incorrect (probably a type). This occurred in the 1860s. But this jump forward 200 years is also problematic, given the focus of the article on the late 1620s/early 1630s.
Ultimately, the author should clarify not only their contribution to the scholarship (e.g. how it is different from Nishimura's work: Since Nishimura has already argued that Feiyin influenced Sessō, the revision should show how this paper shows something different, and why that is important. ), but also how it contributes more broadly to our knowledge. For instance, perhaps they could state that this research demonstrates the complexity and transnational nature of the anti-Christian ideology that became so fundamental to the Tokugawa order. Or something else perhaps.
Overall, very interesting and promising, but with a need for revisions to raise it to the level of good, publishable scholarship.
Author Response
Dear reveiwer,
Thank you for your detailed suggestions.
Please check the attached reply.

Reviewer 3 Report
This paper sets out with a clear agenda and adheres to it throughout. It is important to consider Sesso Sosai's Chinese precedents in terms of his argument against Christianity. I feel the author has successfully demonstrated Sesso's debt to Feiyin and Miyun, among others. Although the author mentions Matteo Ricci and his work _The True Meaning of the Lord of Heaven_ an even greater exposition of this work's role in spurring the production of such texts would have been helpful, but this might be a future project for the author.
The English of the main text is fine, but the footnotes need to be edited. There are grammatical errors that suggest they were not written by a native speaker, so before the paper can be submitted the notes need to be checked and edited. Also, it would be helpful to have citations for mentioned works, such as in note 16 where it mentions qutoes from the _Analects_ and the _Doctrine of the Mean_ but does not specify where in the text they appear. Such additions are always appreciated by a reader.
Author Response
Dear reviewer,
I would like to express my gratitude for your kind review and your precious time spent on it.
I have made improvements in my notes and please feel free to raise any other problem.
I also added some new content, please check the highlighted parts.